# Ag Nanostructures with Spikes on Adhesive Tape as a Flexible Sers-Active Substrate for In Situ Trace Detection of Pesticides on Fruit Skin

**DOI:** 10.3390/nano9121750

**Published:** 2019-12-09

**Authors:** Jaya Sitjar, Jiunn-Der Liao, Han Lee, Li Peng Pan, Bernard Haochih Liu, Wei-en Fu, Guo Dung Chen

**Affiliations:** 1Department of Materials Science and Engineering, National Cheng Kung University, 1 University Road, Tainan 701, Taiwan; jaya.sitjar@gmail.com (J.S.); rick594007@hotmail.com (H.L.); bryan820722@gmail.com (L.P.P.); hcliu@mail.ncku.edu.tw (B.H.L.); 2Medical Device Innovation Center, National Cheng Kung University, 1 University Road, Tainan 701, Taiwan; 3Center for Measurement Standards, Industrial Technology Research Institute, No. 321, Kuang Fu Road, Sec. 2, Hsinchu 300, Taiwan; weienfu@itri.org.tw (W.-e.F.); Eric_chen@itri.org.tw (G.D.C.)

**Keywords:** nanostructures with spikes, surface-enhanced Raman scattering, flexible support, in situ trace detection, pesticides

## Abstract

Nanostructures with spikes (NSPs) have been a subject of several surface-enhanced Raman scattering (SERS) applications owing to their significant Raman signal enhancement brought about by the combined effects of interspike coupling and the accumulated induction on the tips of spikes. Thus, NSPs offer great potential as a SERS-active substrate for relevant applications that require a high density of enhanced “hot spots”. In this study, Ag NSPs were synthesized in varying degrees of agglomeration and were thereafter deposited onto a transparent adhesive tape as a flexible substrate for SERS applications, specifically, in the detection of trace amounts of pesticides. These flexible substrates were referred to as Ag NSPs/tape and optimized with an enhancement factor (EF) of ca. 1.7 × 10^7^. A strong resulting signal enhancement could be attributed to an optimal degree of agglomeration and, consequently, the distances among/between spikes. Long spikes on the synthesized core of Ag NSPs tend to be loosely spaced, which are suitable in detecting relatively large molecules that could access the spaces among the spikes where “hot spots” are generally formed. Since one side of the transparent tape is adhesive, the paste-and-peel off method was successful in obtaining phosmet and carbaryl residues from apple peels as reflected in the acquired SERS spectra. In situ trace detection of the pesticides at low concentrations down to 10^−7^ M could be demonstrated. In situ trace detection of mixed pesticides was possible as the characteristic peaks of both pesticides were observed in equimolar mixtures of the analytes at 10^−2^ to 10^−4^ M. This study is, thus, premised upon applying for in situ trace detection on e.g., fruit skin.

## 1. Introduction

Plasmonic nanostructures (PNSs) exhibit excellent optical properties that allow them to produce intensified localized electric fields near their surfaces as a result of the excitation of their localized surface plasmon resonance (LSPR) mode. This phenomenon provides the capability of PNSs to be utilized as substrates in amplifying characteristic Raman signals of analytes in an ultrasensitive detection technique of surface-enhanced Raman spectroscopy (SERS) [1,2].

The enhancement of Raman signals in SERS is driven by electromagnetic (EM) or chemical (CM) mechanism [3], brought about by the analytes being in close proximity with hot spots located at the gaps among the PNSs, and the electronic properties of the analyte being modified from the formation of metal-analyte complexes, respectively. The contribution by EM to the enhancement effect is more than that of CM, and it is based on the excited state of LSPR as induced by the incident light onto the metal surface, thus this mechanism is dependent on the physical attributes (shape, size, and material/s) of PNSs. On the other hand, the effect from the CM is analyte-dependent as it relies on the transfer of charges that occurs between PNSs and analytes [4,5].

Nanofabrication techniques have already been explored to produce SERS-active substrates in a variety of forms such as colloidal suspensions of nanoparticles (NPs) and custom-made PNSs [6]. It has been reported that among numerous morphologies, PNSs, exhibiting sharp protrusions have the advantage of the lightning rod effect wherein the tips and thin gaps serve as “hot spots” having large electric fields—the tips concentrate the energy in a small and localized volume, leading to the formation of regions of high EM enhancement at the tips of these sharp features [5,7]. Nanostructures with spikes (NSPs) are, thus, highly favored substrates for the advancement of SERS, which casually provide multiple edges from a core that are competent to act as “hot spots” of different levels. However, as these are commonly synthesized by colloidal dispersion, poor reproducibility may occur due to the inhomogeneity of the structures, resulting from uncontrolled aggregation.

Thus, it would be a challenge to implement NSPs as a SERS-active substrate for a state-of-use application. Immobilization of colloidal NPs or NSPs on a flexible support has been developed to address the problems with aggregation simply through deposition onto a solid substrate such as glass and silicon [8]. This may allow solution-synthesized NSs or NSPs to obtain a more reproducible stack structure. In addition, the use of flexible support for NSs or NSPs is further anticipated to widen the applicability of the as-integrated substrate, as they may directly collect target analytes from a surface that is not flat. Examples of NSs or NSPs on a flexible support can be found for the detection of pesticides on e.g., fruit skin or drug molecules on clothes [1]. 

In the fabrication of flexible SERS-active substrates, NSs are first synthesized through wet chemical methods and are then deposited onto a flexible support such as paper [9], polymer strips [1,9,10], or adhesive tapes [11,12] through drop-casting or dip coating. An advantage of flexible support over a conventional rigid one is their capability to perform in situ measurements that would be highly beneficial in monitoring the presence of pesticide residues or organic pollutants on the outer surfaces of agricultural products, for there is no need to perform solvent extraction and invasive sampling [12]. Adhesive tapes have, thus, been used as a flexible support that could capture the analyte molecules on the surface of samples. Despite the convenience that flexible support for SERS substrate has to offer for on-site analyte detection, the complete extraction of analytes on the sample surface, without any assistance from organic solvents, still poses a challenge. However, it should also be considered that the use of a flexible support could result in unwanted background signals and noise, in addition to the incomplete collection of analytes from a sampling surface [13,14].

In this work, a flexible SERS-active substrate was fabricated through drop casting of synthesized silver NSPs (Ag NSPs) onto an adhesive tape. NSPs, which reportedly provide a large Raman signal enhancement, combined with the versatile applicability of a flexible adhesive support, makes an innovative substrate. The aim is therefore to demonstrate a method of rapid analyte sample collection that is suitable for a range of in situ applications, while maintaining high sensitivity. The design of Ag NSPs was found to be suitable not only for relatively small molecules, but also for larger ones that have access to the spaces between/among the spikes where “hot spots” are situated.

## 2. Experimental Section

### 2.1. Synthesis of Surface-Enhanced Raman Scattering (SERS)-Active Silver Nanostructures with Spikes Deposited on Tape (Ag NSPs_x/Tape)

Ag NSPs with an average size of 200~250 nm were prepared by mixing 500 μL of 6 × 10^−2^ M hydroxylamine (NH_2_OH) and 500 μL of 0.05 M of sodium hydroxide (NaOH) to which 45 mL of 10^−3^ M AgNO_3_ was then added and vigorously stirred at 500 rpm for 5 min, as illustrated in Figure 1a. Upon observation of a change in color of the mixture to dark brown, with a pH value of 5.5, 500 μL of 1% w/v trisodium citrate dihydrate (C_6_H_9_Na_3_O_9_) was added and stirred again for another 15 min.

The colloidal suspension was then allowed to stand for 48 h at room temperature before centrifugation at 6000 rpm for 10 min. The supernatant was removed to extract Ag NSPs precipitate that were then dispersed in ultra-pure water. For the simplicity of referring to the substrates, Ag NSPs in varying suspension concentrations were then denoted as Ag NSPs_*x*, where *x* stands for the suspension concentration in M. 

The immobilization of Ag NSPs on a flexible support surface was done through deposition on a clear adhesive tape (Scotch Transparent Tape 500, 3M, Maplewood, MN, USA). As shown in Figure 1b, the synthesized colloidal suspension of Ag NSPs was simply drop-casted onto the adhesive tape that was fixed onto a glass substrate. The suspension was then left to dry at room temperature for 6 h. The resulting flexible substrates were then referred to as Ag NSPs_*x*/tape. 

### 2.2. Morphological Characterization of Ag NSPs_x/Tape

The shape and morphology of Ag NSPs were confirmed through high-resolution thermal field-emission scanning electron microscope (FE-SEM, JSM-7000, JEOL, Tokyo, Japan). Images of Ag NSPs were taken at secondary electron imaging mode with an accelerating voltage of 10 kV. Ag NSPs with varied concentrations were analyzed for the comparison of morphologies and aggregation behavior. All necessary measurements for the sizes and distributions of Ag NSPs were obtained with the use of ImageJ software (version 1.47, National Institutes of Health, Rockville, MD, USA).

### 2.3. Acquisition of SERS Spectra and Enhancement Factor

The as-prepared flexible Ag NSPs_*x*/tape were tested for the evaluation of SERS effect with the use of a Raman spectrometer (Renishaw, Wotton-under-Edge, UK), having an incident power of 3 mW equipped with an air-cooled charge-coupled device (CCD) as the detector. Rhodamine 6G (R6G) was used as the probe molecule, varying concentrations (10^−2^ to 10^−7^ M) of which were dropped onto the flexible Ag NSPs/tape and were left to dry completely before taking SERS measurements, as presented in Figure 1d. The spot size of Raman laser was fixed at Ø = 1 μm; laser wavelengths of 633 and 785 nm^−1^ with gratings at 1800 and 1200 lines/nm, respectively, were used. The acquired Raman spectra were all processed to remove unnecessary background signals and noise and to correct the baseline. 

### 2.4. Application of Flexible Ag NSPs_x/Tape to Pesticide Detection

The flexible Ag NSPs_*x*/tape was then used to detect varying concentrations (10^−2^ to 10^−7^ M) of phosmet or carbaryl (Sigma Aldrich; pesticides in powder form, St. Louis, MO, USA), as well as equimolar mixtures of both pesticides at concentrations of 10^−2^ to 10^−4^ M. The pesticides were dropped as contaminants onto surfaces of washed apple peel samples and were left to dry at room temperature. As illustrated in Figure 1d, the “paste and peel off” method was used to collect the target analytes wherein the sticky side of the flexible adhesive support was pressed onto the apple peel containing analytes. The as-prepared Ag NSPs_*x*/tape were then carefully lifted off and fixed onto a glass substrate for Raman signal collection. 

## 3. Results and Discussion

### 3.1. Surface Characterization and Agglomeration Behavior of Ag NSPs_x/Tape 

The changes in the agglomeration behavior of Ag NSPs with varying suspension concentrations are shown in Figure 2a–d. For Ag NSPs_0.2/tape, as shown in Figure 2d, Ag NSPs appeared to be heavily clumped, creating densely stacked agglomerates that form a cluster. An enlarged image is shown in Figure 2e, showing randomly-oriented Ag NSPs featuring blunt-tipped spikes, and a single Ag NSP could be distinguished in Figure 2f. The extended length of Ag NSP was measured to be around 200 nm, following the average size measurement of randomly-picked Ag NSPs as shown in the size distribution diagram in Appendix A. The size range of the synthesized Ag NSPs holds true regardless of the suspension concentration since the concentration and amounts of Ag NSP precursor reagents were not varied during the synthesis. 

Upon evaluation of the morphologies, it is important to note the agglomeration behavior corresponding to each suspension concentration since it is essential to consider the formation of “hot spots” among NSPs, which can be deduced from the available surface area formed by these nanostructures. In reference to Figure 2a–d, an evolution in the degree of agglomeration can be observed with increasing suspension concentrations. Upon Ag NSPs_0.025/tape, localized mall clusters of Ag NSPs are formed, leaving large intercluster gaps; on the other hand, upon Ag NSPs_0.2/tape, wherein Ag NSPs are heavily clumped, the clusters are rather accumulated in larger areas such that a film is formed. While it is highly preferred that NSPs agglomerate to some degree, heavy agglomeration may not always provide an optimized SERS-active effect coming from a particular NSP. As compared with the case of a study utilizing Au NPs, a decrease in the peak intensities was observed when Au NPs were observed to be heavily clustered [15]. At suspension concentrations of 0.05 and 0.1 M, the degree of agglomeration is intermediate to that of 0.025 and 0.2 M. 

### 3.2. Evaluation of the SERS Effect of Ag NSPs_x/Tape

Ag NSPs_*x*/tape were competent to enhance the measurement of R6G, as shown in Figure 3a,b. The characteristic peak at 1361 cm^−1^ was observed to be the strongest, so this peak was taken as the point of reference in determining an optimized substrate and laser combination. The intensities of the peak at 1361 cm^−1^ under both laser wavelengths were compared in Figure 3c, and the highest intensities were obtained using Ag NSPs_0.05/tape with the 785 nm laser, which was found to be an optimized condition. At Ag NSPs suspension concentrations greater than 0.05 M, the peak intensities gradually decreased. With varying concentrations of R6G, a trend was observed wherein the logarithmic values of R6G concentrations and the intensities of the peak at 1361 cm^−1^ appeared to be a linear calibration plot as shown in Figure 3d. This indicates that the substrate holds a potential in quantitative determination of analyte even with only SERS peak intensity is given, as also observed in some studies [15,16]. 

The peak enhancement brought by Ag NSPs_0.05/tape is due to sufficient amounts of agglomeration that consequently results in optimized interspike distances. On the other hand, the decrease in peak intensities with increasing suspension concentration is due to heavy agglomeration that results in more radiation being scattered by the system, thus less laser energy is available to form “hot spots” [15,16,17]. In addition, it cannot be explicitly concluded that the enhancement comes from resonance effects brought by the laser wavelength and the LSPR of the substrate since the LSPR information was not investigated. However, it is equally important to note that off-resonance enhancement conditions could occur and provide even better enhancement than resonant conditions [18,19]. The EF value, referred to our previous method [20] was calculated to be 1.7 × 10^7^ owing to the optimal agglomeration of Ag NSPs, which is considered typical to currently-developed SERS substrates that were also designed for pesticide detection [3,15,21]. 

### 3.3. Detection of Pesticides from Apple Peel with the Flexible Ag NSPs_x/Tape

A comparison between SERS spectra of standard solutions from the pesticides that were left to dry onto Ag NSPs_0.05/tape and those of the same analytes obtained from contaminated apple peels through the “paste and peel off” method, are shown in Figure 4. The characteristic peaks of Phosmet (606, 653, 1014, 1189, 1260, 1381, 1409, and 1772 cm^−1^) and Carbaryl (713, 1380, and 1582 cm^−1^), as listed in Appendix A [12,22,23,24], are clearly identified in all cases, being the most distinct at 10^−2^ M. Relatively strong peaks of each pesticide, 606, 653, and 1189 cm^−1^ for Phosmet and 1380 and 1582 cm^−1^ for Carbaryl, were still recognizable at lower concentrations, indicating that the limit of detection of both pesticides is close to 10^−6^–10^−7^ M; it is lower than the maximum residue limits of pesticides in food products as set by the World Health Organization (WHO), which is established to be around 10^−4^–10^−5^ [15]. However, this detection limit still remains higher than that obtained by, e.g., utilizing Ag NPs deposited on porous silicon, in which detection limits were competent to go down to 10^−9^–10^−10^ M [21]. The use of the paste and peel off technique with the flexible Ag NSPs could thus be effectively applied for in situ trace detection on fruit skin. 

Nevertheless, the peaks obtained from standard solutions obtained through drop and dry on Ag NSPs_0.05/tape were much sharper than their counterparts in the paste and peel off method. Background fluorescent signals seemed to have caused some of the peaks to appear broader when the analytes were extracted using the paste and peel off method—unwanted analytes or thin layers of the fruit skin might have been included during the peeling process, which has been cited to be a common problem with this technique [12]. Less analyte could have also been collected as the adhesive strength of the tape might not have been enough to take all of the analyte molecules off the surface of the peel. It is also observed that some peaks appear to deviate from the reported values as a consequence of the variation in the orientation and interactions of the molecules with the substrate [20]. In fact, aside from the shifting of the Raman peaks, it has been found that peaks could also broaden, both of which are consequences of when bonds of the functional groups of the analytes are either weakened or strengthened as a result of the variation in the molecular orientations and binding sites [25]. 

The sensitivity of Ag NSPs_0.05/tape was further tested by detecting both Phosmet and Carbaryl in a mixture condition. In Figure 5a, the major characteristic peaks from both pesticides were identified to be present in all the given mixture concentrations, implying that Ag NSPs_0.05/tape is capable for multiplex detection. Furthermore, the sensitivity of the substrate for detecting pesticides in a mixture condition relative to the standard solution was investigated through their peak intensity ratios. The peak intensity ratios obtained were lower in a mixture condition than in standard solution, implying that the substrate is more sensitive when only individual pesticides are detected; this was expected, since in mixture conditions, various analytes would compete to enter “hot spots” and to be detected through the effect of SERS [26,27]. The decrease in the peak intensity ratios from standard solution to a mixture condition is represented through the percentage reduction (%), as reflected in Figure 5b. Both pesticides follow the same trend of having a decrease in the percentage reduction with decreasing mixture concentrations. This could indicate that Ag NSPs_0.05/tape does not exhibit selectivity to either pesticides, given the fact that Ag simply does not have a strong affinity towards any of the constituent functional groups of both pesticides. However, for both pesticides, the reduction of peak intensity is naturally followed by the decrease of concentrations. The optimal condition situated at the lowest given concentration of 10^−4^ M, wherein presumably there is less competition for “hot spot” sites, whereas at higher concentrations, the analyte molecules might have been so dense that the available “hot spots” were unable to effectively accommodate all of the analytes. This suggests that Ag NSPs_0.05/tape could potentially be applied in a non-selective detection for each of the pesticides in a mixture.

### 3.4. Mechanism of SERS upon Ag NSPs_x/Tape

The tips of the spikes in Ag NSPs, as shown in Figure 2f, serve as regions of high local field enhancement as a result of large electric field confinement due to a high concentration of electrons [7,19]. In addition, the intra-spike gaps, defined to be the spaces formed by the spikes on the same single NSP, are also considered to be “hot spot”-rich regions due to their narrow radius of curvature [28]. Aside from the intra-spike gap, “hot spots” formed on a single Ag NSP, the clustering of these nanostructures also plays an important role in contributing to the SERS effect since the distance among the nanostructures is also critical in providing the enhancement of SERS signals. The agglomeration of Ag NSPs should be sufficient enough to create an effective density of “hot spots” in obtaining an optimized SERS enhancement [17]. Considering the effects of each signal contributing to single or overlapped multiple Ag NSPs, larger regions of high electrical field are created and, consequently, the overall effect of SERS are generated.

Furthermore, the distances among Ag NSPs should also be considered, which could mostly be related to the degree of NSPs agglomeration; it is noted that Ag NSPs with short interparticle distances exhibit a high degree of agglomeration, such as in Figure 2c,d. A proposed illustration of “hot spots” formed between Ag NSPs with respect to interparticle distance and configuration is shown in Figure 6, in which four cases are enumerated. At conditions as shown in Figure 6-(1), an optimal gap distance is needed to form strong large-volume “hot spots”, but this case is the rarest as the optimal interspike distance could only range a few nm. An interspike distance that is higher than the optimum may not be able to create strong “hot spots”, since an interparticle distance of 5 nm is said to be the upper limit for the LSPR to provide SERS enhancement [29]. A tip-to-tip configuration but with the absence of a gap, as illustrated in Figure 6-(2), also produces a strong “hot spot”, but with a volume lower than that in case (1) [30]. Both cases (1) and (2) have the least chances of occurring as Ag NSPs tips need to be in close proximity with each other which could be achieved by increasing the degree of agglomeration. Case (3) involves the tip being in contact with the stem of the spike—this would result to a “hot spot” weaker than that in cases (1) and (2), since the concentration of electrons along the stem of a spike is much less than that in the tip. The last case (4) provides the weakest hot spot due to the interspike gap being too large such that electric fields of the spikes do not overlap sufficiently to produce effective “hot spots”. In the case of sharp-tipped spikes, the formation of “hot spots” with respect to all four given cases is the same as that of the blunt spikes. However, the first two cases, both involving tip-to-tip configurations, are even less likely to occur in sharp-tipped spikes than in blunt ones, since the area of contact with sharper tips is much smaller than that of the blunt tips. Thus, with sharp-tipped Ag NSPs, the tip-to-stem configuration would be the most common case to occur. 

On the other hand, when a single Ag NSP is considered, as illustrated in Figure 2f, it can be observed that its structure consists of a solid core with blunt spikes radially branching out. This particular structure provides an optimized electric field enhancement as charges tend to concentrate on the tip-end of the spikes due to the lightning rod effect, and the addition of a spherical core supplements the effect, since it acts as a reservoir of electrons—allowing more electrons to move from the core to the tips, consequently leading to a stronger electric field enhancement [30]. Both the coupling of Ag NSPs and the lightning rod effect as presented in Figure 6 combine to produce a synergistic effect leading to a strong enhancement of electric fields and, thereafter, to increase in the “hot-spot” volume [30,31,32]. The introduction of spikes thus amplified the SERS effect, particularly the EM mechanism. 

### 3.5. Effect of Ag NSP Spike Length on Target Detection through SERS

Nanostructures exhibiting longitudinal geometries in multiple directions i.e., spiky nanostructures, have been found to effectively allow more light absorption due to intra-spike and interspike gaps [29,30]. However, it is also noted that an optimal length should be taken into consideration—in the case of Ag NSPs, e.g., the local electromagnetic field weakens as it moves from the core of the structure to its tip, leading to weaker SERS signals with spikes longer than the optimal length [33]. In addition, these spiky nanostructures present an advantage over the commonly-used nanospheres as they provide a larger specific surface area that contributes to the enhancement of signals [34]; spiky nanostructures take advantage of the lightning rod effect in providing highly enhanced electric fields [35]. 

The NSPs used in this study feature a few long, outward blunt-tipped spikes around a spherical core, as illustrated in Figure 7a, in contrast to that in Figure 7b wherein one NSP possesses many, short blunt spikes. Self-“hot spots” are generated when a single NSP is being considered—these are situated at the point of attachment of each spike to the core. The overlapping electric fields of these areas lead to the formation of the said “hot spot”, but as these self-“hot spots” are intra-NSPs hot spots, its strength cannot be definitely compared to the four inter-NSP cases (Figure 6). However, it is important to note that when Ag NSPs are assembled in a substrate, all of these 5 cases are present.

In view of the target molecule size with respect to the size of NSP, it is noted that large molecules are able to access the relatively larger gaps between the spikes of Ag NSPs in Figure 7a; these large molecules, however, could not access the self-“hot spots” formed by NSPs in Figure 7b. Hence, to maximize the overall “hot-spot” effect of NSPs, the size of the gaps between the spikes where self-“hot spots” are formed should be sufficient to accommodate the molecule/s of interest.

In applications requiring the detection of relatively large molecules, it is essential to consider the size of the target molecule to design a suitable structure of NSP accordingly. In the case of this study, the detection of large molecules such as pesticides is attributed to the relatively large intra-spike gaps where hot spots are present, which the said molecules could easily have access to, and this would mean that smaller molecules are then anticipated to be detected easily.

## 4. Conclusions

In this study, Ag NSPs measuring around 200 nm in an extended length and featuring blunt tips were successfully synthesized, were prepared in varying degrees of agglomeration, and were made as flexible SERS substrates by depositing these on a transparent tape. Evaluation of the substrate for SERS signal enhancement resulted to an EF of 1.7 × 10^7^, sufficiently sensitive to non-selectively detect pesticides at 10^−7^ M, both in each standard solution and mixture conditions. In addition, the paste and peel method, which was performed to test the effectiveness of the substrate being flexible, showed that pesticides even at a concentration of 10^−7^ are still distinguishable despite the background signal and noise brought by the presence of the transparent tape. Different cases of Ag NSP tip and body configurations in forming “hot spots” were also noted as the mechanism behind the resulting overall SERS effect. These cases varied in the chances of occurrence and strength of formed “hot spots”. 

## Figures and Tables

**Figure 1 nanomaterials-09-01750-f001:**
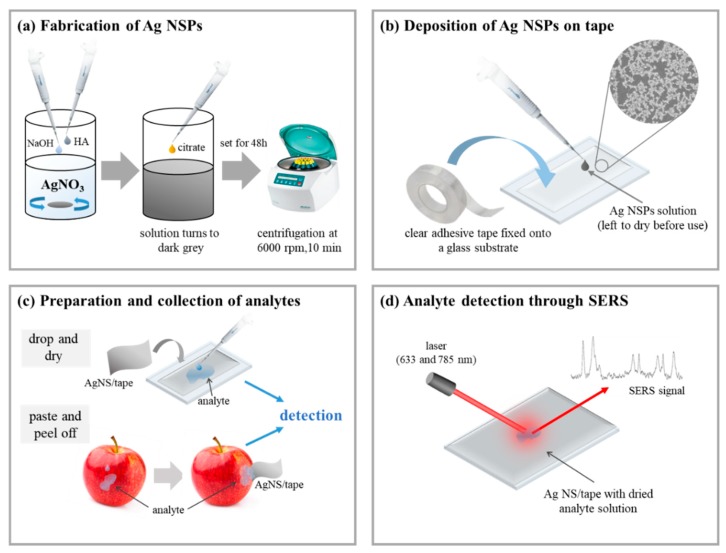
Scheme of experimental procedure: (**a**) synthesis of silver nanostructures with spikes (Ag NSPs) through the reduction of Ag by hydroxylamine and citrate; (**b**) deposition of the synthesized Ag NSPs on to tape; (**c**) collection of analytes through two different methods—the conventional drop and dry of standard analyte solutions and paste and peel off which is basically an in situ collection technique; (**d**) detection of analytes through surface-enhanced raman scattering (SERS) with the use of 633 and 785 nm laser wavelengths.

**Figure 2 nanomaterials-09-01750-f002:**
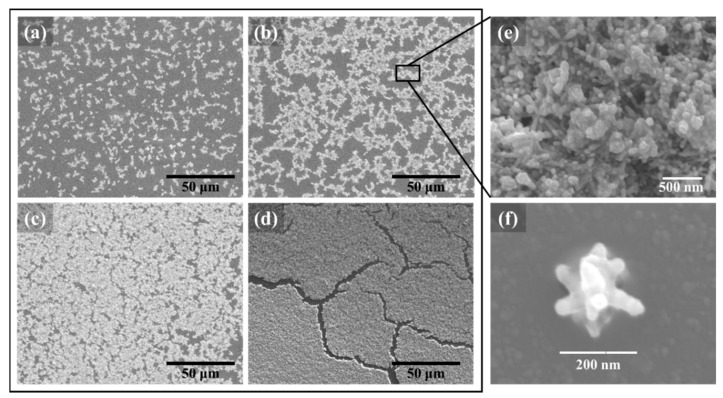
Distribution of varying concentrations of Ag NSPs on tape at (**a**) 0.025 M, (**b**) 0.05 M, (**c**) 0.1 M, and (**d**) 0.2 M; (**e**) close-up image of the agglomeration of Ag NSPs at 0.05 M; (**f**) single Ag NSPs measuring at roughly 200 nm.

**Figure 3 nanomaterials-09-01750-f003:**
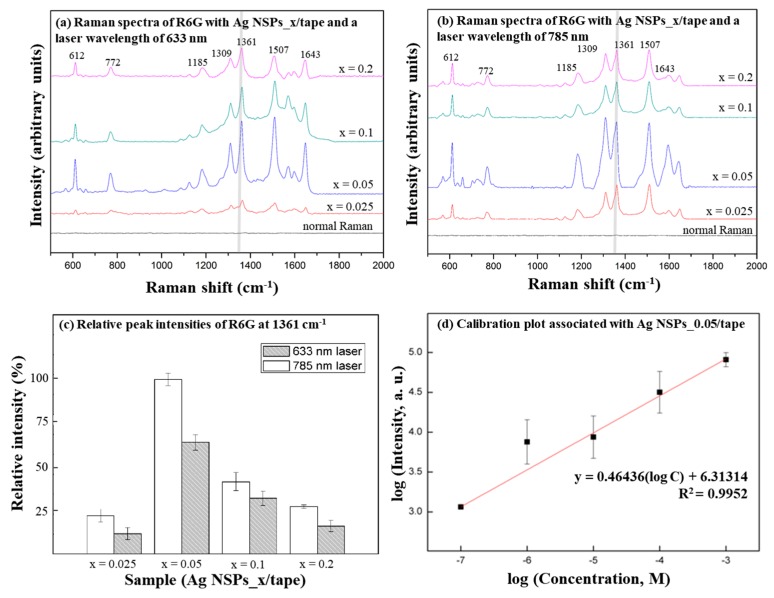
SERS spectra of 10^−2^ M R6G measured at the respective excitation wavelengths of (**a**) 633 nm and (**b**) 785 nm on AgNS solutions at varying concentrations; (**c**) relative Raman intensities of 10^−2^ M R6G at the 1361 cm^−1^ peak (C–C stretching) obtained with the Ag NSPs_*x*/tape at 633 and 785 nm excitation wavelengths exhibit differences in the response of the substrate due to varying degree of agglomeration; (**d**) linear calibration plot produced from the logarithms of the obtained intensities at the 1361 cm^−1^ peak with varying concentrations of R6G using Ag NSPs_0.05/tape.

**Figure 4 nanomaterials-09-01750-f004:**
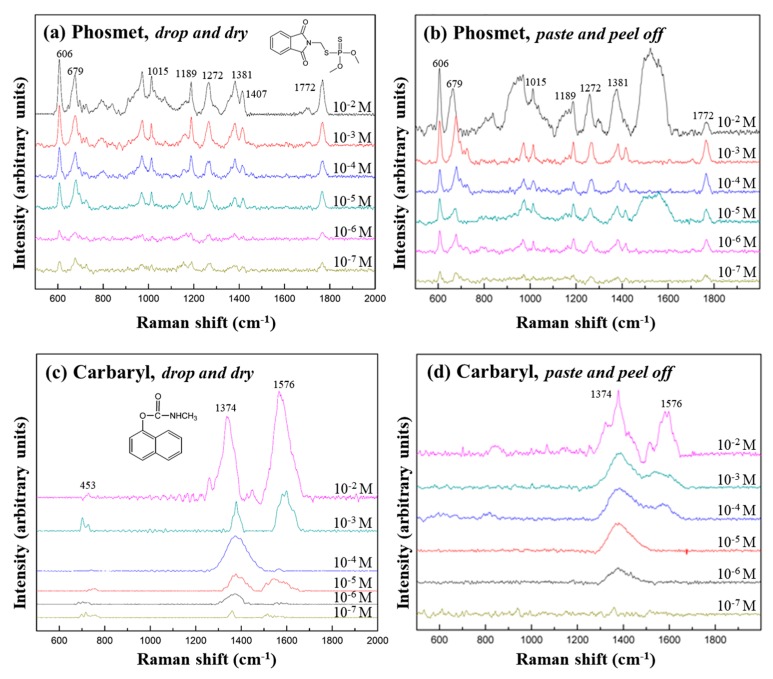
SERS spectra of varying concentrations of pesticides by the (**a**,**c**) conventional drop and dry and (**b**,**d**) paste and peel off method using the 785 nm excitation wavelength laser and AgNSPs_0.05/tape.

**Figure 5 nanomaterials-09-01750-f005:**
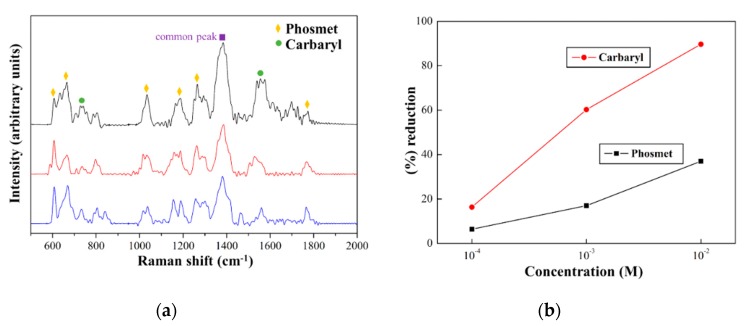
SERS spectra of mixtures of the same molar concentrations of Phosmet and Carbaryl on AgNS_0.05 demonstrate that the substrate is capable of detecting multiple analytes on a single measurement.

**Figure 6 nanomaterials-09-01750-f006:**
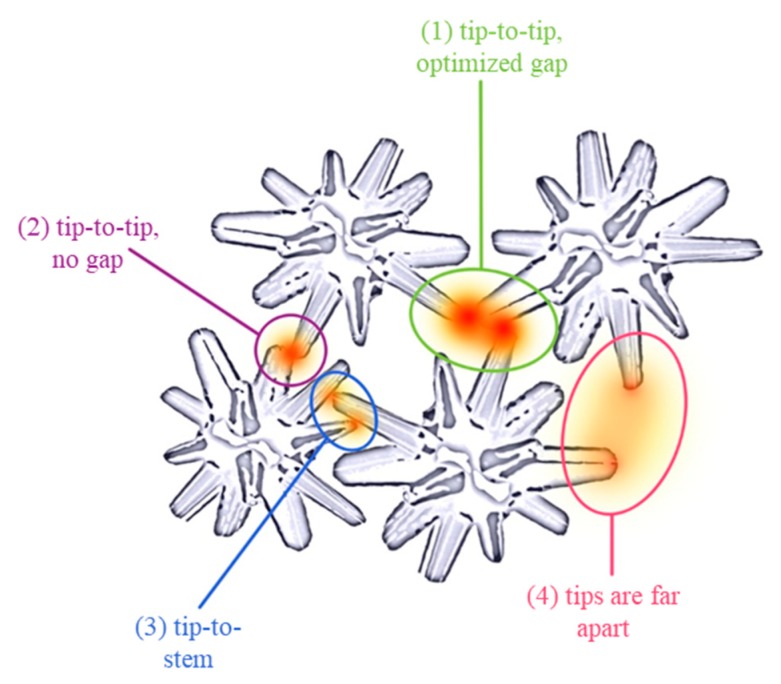
Four cases of Ag NSPs tip configurations leading to the formation of effective “hot spots” decreasing hot spot volume, strength, and rarity of occurrence (rarest to most common) from (1) to (4); (1) tip-to-tip configuration with an optimal gap, which provides the strongest “hot spot” since the tips are separated at such distance that gives out the largest “hot-spot” volume without compromising strength; (2) tip-to-tip with no gap leaves no space for a larger “hot spot” to be formed but provides a “hot spot” that is slightly weaker than that of the first case; (3) tip-to-stem provides a weaker “hot spot” since only one tip is involved compared to the first 2 configurations; and lastly, (4) tips that are distanced far apart forms very weak “hot spots” since the electromagnetic fields are not sufficiently close to form significantly strong “hot spots”.

**Figure 7 nanomaterials-09-01750-f007:**
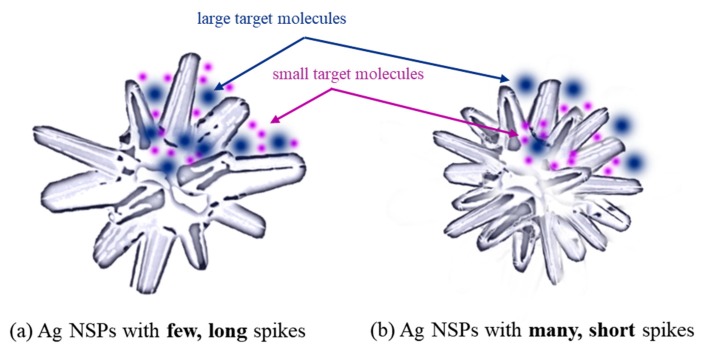
A comparison of the positioning of analytes in nanostructures with spikes of different lengths: (**a**) few long spikes around a nanosphere core create high-volume “hot spots” at the gaps created between the long spikes that are easily accessible for both large and small target molecules; (**b**) many short spikes around a nanosphere core create low-volume “hot spots” and are suitable in detecting analytes that are small enough to place themselves in between the spikes and “hot spots”.

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
