# Peer review of "Ag Nanostructures with Spikes on Adhesive Tape as a Flexible Sers-Active Substrate for In Situ Trace Detection of Pesticides on Fruit Skin"

_nanomaterials, 2019, doi:10.3390/nano9121750_

Round 1
Reviewer 1 Report
Nanomaterials – 660324
The manuscript entitle “Ag nanostructures with spikes on adhesive tape as a flexible SERS-active substrate for in-situ trace detection of pesticides on fruit skin” submitted by Sitjar et al. reports the preparation of active spiked Ag NPs deposited on tap for the SERS detection of pesticides. However these kind of SERS substrates are not new, the work was well performed and the results are clear and well explained.
In general, the manuscript fits very well into the field covered by the journal Nanomaterials. However, the current state of the manuscript is below the journal standard. Consequently, I recommend publication in Nanomaterials after major revision.
Abstract, line 24, page 1: what analyte?? the authors should be more specific. Can the authors present the UV-Vis spectra for the Ag NPs substrates with different concentration of particles. I would be interesting see the surface plasmon resonance (SPR) band of the substrate in function of the agglomeration state. How the shift of the SPR can influence the Raman measurements?
3. Page5, line 174: Why the authors observed a higher enhancement of the R6G Raman signal using the laser 785 nm instead of 633nm?
4 Why the Raman spectrum of R6G is different if the authors use the 785 nm and the 633nm as excitation laser? It is observed differences in the intensities of the bands between 1500-1700 cm-1 and 1300-1400 cm-1 (Figure 3a and b)
5. I do not understand the section 3.5. How the authors correlate their results with the effect of Ag NSP spike length? The small target and large targets can b compared to the pesticides used in this work?
Other observations
-3. page 5, line 167; 0.5M or 0.2M?
- I believe that Figure 7 should be Figure 5
Author Response
Nanomaterials
Ms. Ref. No.: nanomaterials-660324
Title: Ag nanostructures with spikes on adhesive tape as a flexible SERS-active substrate for in-situ trace detection of pesticides on fruit skin
Dear Editor,
We highly appreciate the reviewers’ feedback and suggestions to improve the quality of the manuscript. Their addressed questions have been carefully replied. Attached please find the following two files: the replies to reviewers with a list of changes and the revised manuscript.
We would also like to note that in our manuscript, Figure S1 and Table S1 seemed to appear in our main text but we have separately uploaded the file for the supplementary data.
Thank you very much for your kind consideration to publish this paper in a regular issue.
Yours sincerely,
Prof. Dr. Jiunn-Der Liao
The corresponding author
Replies to reviewers
Reviewer #1: (Q: Question, R: Reply)
(Q#1) Abstract, line 24, page 1: What analyte?? The authors should be more specific.
(R#1) Thank you for the reviewer’s suggestions, and we have taken these into consideration and made the necessary changes in the latest manuscript revision. We have stated the particular application of our substrate in line 24 of the Abstract; we have marked the changes in red:
“…for SERS applications, in the detection of trace amounts of pesticides to be specific.”
(Q#2) Can the authors present the UV-Vis spectra for the Ag NPs substrates with different concentration of particles. I would be interesting see the surface plasmon resonance (SPR) band of the substrate in function of the agglomeration state. How the shift of the SPR can influence the Raman measurements?
(R#2) Thank you for addressing this option. We have actually considered this but for future investigations, so we cannot provide UV-Vis spectra results for now as this was not under the scope of the current study. However, we gladly take note of your remarks and would apply this in our further studies.
(Q#3) Page 5, line 174: Why the authors observed a higher enhancement of the R6G Raman signal using the laser 785 nm instead of 633nm?
(R#3) As discussed in our previous paper [1], the choice of laser’s wavelength is quite dependent on the synergetic effect among nanostructures, analytes, and the applied Raman laser. However, as also discussed [1, 2], regarding to the SPR band, the resonant wavelength might be possible to enhance the R6G signal, which is higher at 785 nm than 633 nm since the resonant wavelength might be at or close to 785 nm.
Sitjar, J.; Liao, J.-D.; Lee, H.; Liu, B.H.; Fu, W. SERS-Active Substrate with Collective Amplification Design for Trace Analysis of Pesticides. Nanomaterials 2019, 9, 664. Li, A.; Srivastava, S.K.; Abdulhalim, I.; Li, S. Engineering the hot spots in squared arrays of gold nanoparticles on a silver film. Nanoscale 2016, 58, 267–297.
(Q#4) Why the Raman spectrum of R6G is different if the authors use the 785 nm and the 633nm as excitation laser? It is observed differences in the intensities of the bands between 1500-1700 cm-1 and 1300-1400 cm-1 (Figure 3a and b)
(R#4) As also mentioned in R#3, the changes in the spectra obtained with 633 nm and 785 nm could be caused by the resonant wavelength possibly being at or close to 785 nm. In addition, the broadening and shifting of peaks normally occur due to the variation of molecular orientations and binding sites. This has in fact been briefly stated in lines 226-229 in the latest revision of the manuscript. The changes are marked in red:
“… a consequence of the variation in the orientation and interactions of the molecules with the substrate [19]. In fact, aside from the shifting of the Raman peaks, it has been found that peaks could also broaden, both of which are consequences when bonds of the functional groups of the analytes are either weakened or strengthened, as a result of the variation in the molecular orientations and binding sites [24].”
(Q#5) I do not understand the section 3.5. How the authors correlate their results with the effect of Ag NSP spike length? The small target and large targets can be compared to the pesticides used in this work?
(R#5) Thank you for this comment. We included section 3.5 as we wanted to have an extensive discussion on the structure produced in this study, which in this case is the specific structure of Ag NSP. In section 3.4, various cases of Ag NSP tip configurations and the resulting hot spots were discussed, and were referred to as interspike hot spots.
However, aside from these, intra-spike hot spots also exist and we wanted to have a separate discussion for it since analyte size plays an important role in the effectivity of this kind of hot spot. This section is mainly our assumption and discussion of the possible mechanism/s surrounding the effect of intraspike “length and spacing” on the positioning of analyte molecules on the nanostructures.
(Q#6) page 5, line 167; 0.5M or 0.2M?
(R#6) Thank you for pointing this out. We have corrected this and marked the changes in red, as reflected in page 5, line 168.
(Q#7) I believe that Figure 7 should be Figure 5
(R#7) Thank you for pointing this out. We have checked the manuscript and it seems that we have mislabeled the figure numbers. We have made all the necessary changes and reflected these on the latest revision.
Reviewer #2: (Q: Question, R: Reply)
(Q#1) In Fig. 3c the authors represent the relative intensity of R6G peaks at different Raman excitation wavelengths. The figure should show values on the y-axis and the authors should clarify the reference point for such relative intensities. Also, the authors should clarify how they compare intensities taken at different excitations. Finally, the authors may optionally consider a scatter plot for this figure, instead of bar chart, for clarity purpose.
(R#1) Thank you for the suggestions. We have made the necessary corrections in the said figure and have shown the values on the y-axis, which are the relative intensities expressed in %. The relative intensities were obtained by taking the highest intensity value at the 1361 cm-1 peak among the samples and excitation wavelengths (in this case, at x = 0.05 with the 785 nm laser) and correspondingly setting this as 100%. Through this means, it would be easier to compare the intensities obtained with varying samples and laser wavelengths, thus we have opted to use the bar chart. We would gladly take into your consideration of presenting the results in the form of a scatter plot for our future studies.
(Q#2) Fig. 5 shows Raman spectra taken at different concentrations for two different analytes in both drop and dry and paste and peel off experiments. It seems that results for phosmet are a lot more promising than for carbaryl as the peaks are visible in both experiment types down to very low concentration, while for carbaryl the peaks are significantly affected by the experiment. The author should discuss this point as well as also comment on the fact that in some instances the 1576 cm-1 peak seems to disappear.
(R#2) Thank you for bringing this up. As for the 1576 cm-1 peak disappearing with decreasing concentration, it could possibly be caused by the gradual weakening of the C=C stretching (vibrational mode corresponding to the 1576 cm-1 peak). As for the differences in phosmet and carbaryl, we think that for phosmet, the peaks appear to be stronger because it might be that the carbaryl has only a few characteristic peaks compared to phosmet so we think we cannot make a definite comparison for the pesticides, but it would be interesting to consider your remark to further investigate on the possible selective properties of the substrate.
(Q#3) The authors should also articulate the point expressed in lines 223-225 in the interest of more clarity for the reader.
(R#3) Thank you for this suggestion. We have added another reference that clearly explains the possible reason as to why peak shifting happen and included this in lines 226-229. We have marked the changes in red.
“… a consequence of the variation in the orientation and interactions of the molecules with the substrate [19]. In fact, aside from the shifting of the Raman peaks, it has been found that peaks could also broaden, both of which are consequences when bonds of the functional groups of the analytes are either weakened or strengthened, as a result of the variation in the molecular orientations and binding sites [24].”
(Q#4) The authors should redefine figures as fig. 4 seem to appear after fig. 6 and before fig. 7 In fig. 6b the x-axis seems wrong as Raman shift does not make sense to me. It probably is concentration
(R#4) Thank you for bringing this up. We have already made the necessary corrections regarding this issue as the figures were wrongly labelled. The x-axis in Figure 6b has also been corrected ̶ it should be “concentration” instead of Raman shift.
(Q#5) The consideration expressed in lines 305-307 should be supported by appropriate referencing.
(R#5) Thank you for your comments. We have made the appropriate changes and have included the references we used for this statement.
(Q#6) The statement made in lines 333-335 should be more articulated and supported, maybe referencing to the data reported in the article or in a way that seems appropriate to the authors
(R#6) Thank you for this suggestion. We have reflected the necessary changes in relation to this in our manuscript. The changes in lines 335-336 are marked in red as shown below:
“…… relatively large intra-spike gaps where “hot spots” are present, which the said molecules could easily have access to, and this would…”
(Q#7) The labels “_x” and “_x/tape” are made explicit in lines 105-106 and 111, but they appear earlier in the text and also in the abstract. I would avoid its use in the abstract and in the text before it is made explicit Line 23: were instead of was
(R#7) Thank you for pointing this out. We have decided to instead state in the abstract what Ag NSPs/tape refers to. Also in the abstract, we have also removed the value “0.05” in Ag NSPs_0.05/tape. We have made the necessary changes, marked in red as shown:
“…… These flexible substrates were referred to as Ag NSPs/tape and was optimized with an enhancement factor (EF) of …”
List of changes
|
# |
Page /Line |
Original |
Revised |
|
1 |
1/24 |
…for SERS applications. Ag NSPs_0.05/tape… |
…for SERS applications, in the detection of trace amounts of pesticides to be specific. Ag NSPs_0.05/tape… |
|
2 |
5/168 |
…intermediate to that of 0.025 and 0.5 M. |
…intermediate to that of 0.025 and 0.2 M. |
|
3 |
10/307-308 |
… due to intra-spike and interspike gaps. |
…due to intra-spike and interspike gaps [29, 30]. |
|
4 |
11/335-336 |
… relatively large intra-spike gaps and this would… |
… relatively large intra-spike gaps where hot spots are present, which the said molecules could easily have access to, and this would… |
|
5 |
8/226-229 |
…a consequence of the variation in the orientation and interactions of the molecules with the substrate [19]. |
…a consequence of the variation in the orientation and interactions of the molecules with the substrate [19]. In fact, aside from the shifting of the Raman peaks, it has been found that peaks could also broaden, both of which are consequences when bonds of the functional groups of the analytes are either weakened or strengthened, as a result of the variation in the molecular orientations and binding sites [24]. |
|
6 |
1/23 |
…agglomeration and was thereafter deposited onto a transparent adhesive… |
…agglomeration and were thereafter deposited onto a transparent adhesive… |
|
7 |
1/24-25 |
… Ag NSPs_0.05/tape was optimized… |
…These flexible substrates were referred to as Ag NSPs/tape and was optimized… |

Reviewer 2 Report
The paper “Ag nanostructures with spikes on adhesive tape as a flexible SERS-active substrate for in-situ trace detection of pesticides on fruit skin” describes the characterization of silver nanostructures and their application as pesticide sensors through SERS effect, exploiting the branched structure of such materials that creates hotspots.
The article is well written and the results are interesting so I recommend publication on the journal Nanomaterials, as long as the following points are addressed:
In Fig. 3c the authors represent the relative intensity of R6G peaks at different Raman excitation wavelengths. The figure should show values on the y-axis and the authors should clarify the reference point for such relative intensities. Also, the authors should clarify how they compare intensities taken at different excitations. Finally, the authors may optionally consider a scatter plot for this figure, instead of bar chart, for clarity purpose. Fig. 5 shows Raman spectra taken at different concentrations for two different analytes in both drop and dry and paste and peel off experiments. It seems that results for phosmet are a lot more promising than for carbaryl as the peaks are visible in both experiment types down to very low concentration, while for carbaryl the peaks are significantly affected by the experiment. The author should discuss this point as well as also comment on the fact that in some instances the 1576 cm-1 peak seems to disappear. The authors should also articulate the point expressed in lines 223-225 in the interest of more clarity for the reader. The authors should redefine figures as fig. 4 seem to appear after fig. 6 and before fig. 7 In fig. 6b the x-axis seems wrong as Raman shift does not make sense to me. It probably is concentration The consideration expressed in lines 305-307 should be supported by appropriate referencing. The statement made in lines 333-335 should be more articulated and supported, maybe referencing to the data reported in the article or in a way that seems appropriate to the authors The labels “_x” and “-x/tape” are made explicit in lines 105-106 and 111, but they appear earlier in the text and also in the abstract. I would avoid its use in the abstract and in the text before it is made explicit Line 23: were instead of was
Author Response

(The authors gave the same response as above.)

Round 2
Reviewer 1 Report
The authors answered all of the posed comments. However, not all issues have been properly resolved and some open questions remain. I cannot recommend publication in Nanomaterials before unless further revisions. Below, I repeat comment #2 and #3 which was not answered satisfactorily. 
Comments:
1) In the earlier report, the authors were asked to comment on the surface plasmon resonance (SPR) band of the substrate in function of the agglomeration state and the effect of resonance conditions of the analyte.
First, "Does the laser line coincide with the LSPR peak of the substrate? If not, this could be explained by off-resonant enhancement. Compare "Plasmonics in Sensing: From Colorimetry to SERS Analytics", DOI: 10.5772/intechopen.79055."
How the authors can say “regarding to the SPR band, the resonant wavelength might be possible to enhance the R6G signal, which is higher at 785 nm than 633 nm since the resonant wavelength might be at or close to 785 nm.” if the R6G is not in resonance with the laser used either 633nm or 785nm (maximum absorbance at 535 nm) and they do not present any absorbance spectrum of at least one of the Ag substrates.
Author Response
Nanomaterials
Ms. Ref. No.: nanomaterials-660324
Title: Ag nanostructures with spikes on adhesive tape as a flexible SERS-active substrate for in-situ trace detection of pesticides on fruit skin
Dear Editor,
We highly appreciate the reviewers’ feedback and suggestions to improve the quality of the manuscript. Their addressed questions have been carefully replied. Attached please find the following two files: the replies to reviewers with a list of changes and the secondly revised manuscript.
We would also like to note that in our manuscript, Figure S1 and Table S1 seemed to appear in our main text but we have separately uploaded the file for the supplementary data.
Thank you very much for your kind consideration to publish this paper in a regular issue.
Yours sincerely,
Prof. Dr. Jiunn-Der Liao
The corresponding author
Replies to reviewers
Reviewer #1: (Q: Question, R: Reply)
In the earlier report, the authors were asked to comment on the surface plasmon resonance (SPR) band of the substrate in function of the agglomeration state and the effect of resonance conditions of the analyte.
(Q#1) First, "Does the laser line coincide with the LSPR peak of the substrate? If not, this could be explained by off-resonant enhancement. Compare "Plasmonics in Sensing: From Colorimetry to SERS Analytics", DOI: 10.5772/intechopen.79055."
(Q#1) Thank you for your suggestion. We took into consideration the paper you recommended and found that off-resonance enhancement could possibly explain the resulting SERS enhancement in the system despite the LSPR peak does not coincide with the laser wavelength. But since we do not have any data providing the LSPR peak of the substrate, we cannot explicitly state that the enhancement caused by the laser wavelengths used are solely coming from off-resonance enhancement. With this, we made some changes in our manuscript stating that off-resonance enhancement could be possible in the case of our study. The changes in lines 193-196 are marked in red:
“…“hot spots” [15, 16]. In addition, it cannot be explicitly concluded that the enhancement comes from resonance effects brought by the laser wavelength and the LSPR of the substrate since the LSPR information was not investigated. However, it is equally important to note that off-resonance enhancement conditions could occur and provide even better enhancement than resonant conditions [19].”
(Q#2) How the authors can say “regarding to the SPR band, the resonant wavelength might be possible to enhance the R6G signal, which is higher at 785 nm than 633 nm since the resonant wavelength might be at or close to 785 nm.” if the R6G is not in resonance with the laser used either 633nm or 785nm (maximum absorbance at 535 nm) and they do not present any absorbance spectrum of at least one of the Ag substrates.
(R#2) Thank you for your comment. With regards to this, since we cannot provide any information about the LSPR of both the analyte and the substrate, we did not include any discussion in relation to the resonance effects of the LSPR. However, since a recommended reference mentioned in Q#1 may be significant for our study, we thus decide to cite the said paper and state that “… However, it is equally important to note that off-resonance enhancement conditions could occur and provide even better enhancement than resonant conditions [19].”
List of changes
|
# |
Page /Line |
Original |
Revised |
|
1 |
1/23 |
…agglomeration and was thereafter deposited onto a transparent adhesive… |
…agglomeration and were thereafter deposited onto a transparent adhesive… |
|
2 |
1/24-25 |
…for SERS applications. Ag NSPs_0.05/tape… |
…for SERS applications, in the detection of trace amounts of pesticides to be specific. These flexible substrates were referred to as Ag NSPs/tape and was optimized… |
|
3 |
5/168-169 |
…intermediate to that of 0.025 and 0.5 M. |
…intermediate to that of 0.025 and 0.2 M. |
|
4 |
6/193-197 |
…“hot spots” [15, 16]. The EF value, referred to our previous method… |
…“hot spots” [15, 16]. In addition, it cannot be explicitly concluded that the enhancement comes from resonance effects brought by the laser wavelength and the LSPR of the substrate since the LSPR information was not investigated. However, it is equally important to note that off-resonance enhancement conditions could occur and provide even better enhancement than resonant conditions [19]. The EF value, referred to our previous method… |
|
5 |
8/231-234 |
…a consequence of the variation in the orientation and interactions of the molecules with the substrate [19]. |
…a consequence of the variation in the orientation and interactions of the molecules with the substrate [19]. In fact, aside from the shifting of the Raman peaks, it has been found that peaks could also broaden, both of which are consequences when bonds of the functional groups of the analytes are either weakened or strengthened, as a result of the variation in the molecular orientations and binding sites [24]. |
|
6 |
10/316 |
… due to intra-spike and interspike gaps. |
…due to intra-spike and interspike gaps [29, 30]. |
|
7 |
11/344 |
… relatively large intra-spike gaps and this would… |
… relatively large intra-spike gaps where hot spots are present, which the said molecules could easily have access to, and this would… |

Round 3
Reviewer 1 Report
The manuscript can be accept to be publish in Nanomaterials in the present form